# Efficacy and Immunomodulating Properties of Eltrombopag in Aplastic Anemia following Autologous Stem Cell Transplant: Case Report and Review of the Literature

**DOI:** 10.3390/ph15040419

**Published:** 2022-03-30

**Authors:** Marta Bortolotti, Loredana Pettine, Anna Zaninoni, Giorgio Alberto Croci, Wilma Barcellini, Bruno Fattizzo

**Affiliations:** 1Fondazione IRCCS Ca’ Granda Ospedale Maggiore Policlinico, 20122 Milan, Italy; marta.bortolotti@unimi.it (M.B.); loredana.pettine@policlinico.mi.it (L.P.); anna.zaninoni@policlinico.mi.it (A.Z.); giorgio.croci@policlinico.mi.it (G.A.C.); wilma.barcellini@policlinico.mi.it (W.B.); 2Department of Oncology and Hemato-Oncology, University of Milan, 20122 Milan, Italy

**Keywords:** eltrombopag, aplastic anemia, autologous transplant, immunomodulation

## Abstract

Thrombopoietin receptor agonists (TPO-RA) are currently indicated for the treatment of chronic immune thrombocytopenia and relapsed refractory aplastic anemia. However, the off-label use of these drugs is more and more frequent, including in the setting of aplasia secondary to chemotherapy and hemopoietic stem cell transplant (SCT). Growing evidence suggests that mechanisms of action of TPO-RA go beyond the TPO-receptor stimulation and point at the immunomodulating properties of these drugs. Here, we present a case of prolonged bone marrow aplasia secondary to autologous SCT treated with eltrombopag. We describe the clinical efficacy and the immunomodulating effect of this drug on inflammatory cytokine profile and bone marrow histology. Furthermore, we provide a review of the most recent literature highlighting the efficacy and safety of TPO-RA after SCT and chemotherapy for hematologic conditions.

## 1. Introduction

Thrombopoietin receptor agonists (TPO-RA), eltrombopag and romiplostim, are two polyhedric drugs initially licensed for the treatment of immune thrombocytopenia (ITP). Both act by triggering the TPO receptor on megakaryocytes (MK) and hematopoietic stem cells, inducing activation of the JAK2/STAT5 pathway with subsequent MK proliferation and an increase in platelet generation [1,2]. However, the two drugs present some differences both in biological properties and mechanisms of action [3]. In particular, eltrombopag appears to target the earlier precursor of MK, stimulating not only the proliferation pathway but also differentiation [1,4]. Furthermore, growing evidence is pointing at a broader spectrum of action of this small molecule, including immunomodulating properties and iron mobilization [5]. Eltrombopag is administered orally and is currently indicated for the treatment of chronic ITP after failure of first line therapy, relapsed refractory aplastic anemia (AA) and thrombocytopenia secondary to hepatitis C during treatment with interferon (IFN). Romiplostim has a subcutaneous route of administration and is licensed for chronic ITP only. However, the off-label use of these two drugs is more and more frequent, particularly in the setting of aplasia secondary to chemotherapy and hemopoietic stem cell transplant (HSCT), and thrombocytopenia associated with lymphoproliferative disorders and myelodysplastic syndromes (MDS) [6]. In this case report, we describe the clinical efficacy and the immunomodulating effect of eltrombopag on inflammatory cytokine profile in a patient experiencing prolonged aplasia following autologous stem cells transplant (ASCT) for amyloidosis AL. Furthermore, we provide a review of the most recent literature highlighting the efficacy of TPO-RA after HSCT and chemotherapy for hematologic conditions.

## 2. Methods

The patient gave informed consent, and the study was conducted according to the Helsinki declaration. All information regarding patient clinical history, hematologic parameters and bone marrow (BM) evaluation were prospectively collected from September 2019 until the time of writing. Cytokine studies and molecular studies by next generation sequencing (NGS) were sequentially performed during eltrombopag therapy. The following cytokines were evaluated in serum using commercial ELISA kits (High Sensitivity Elisa kits, Invitrogen by Thermo Fisher Scientific, Boston, MA, USA, human TGF-beta elisa kit, Immunological Sciences, Rome, Italy): interleukin (IL)6, IL10, IL17, tumor necrosis factor (TNF)-alpha, interferon (IFN)-gamma, and transforming growth factor (TGF)-beta. Cytokine levels were compared with 40 age- and sex-matched healthy controls. The NGS study was performed by next generation sequence technology (Ion Torrents5), Ion Reporters software 5.2, which evaluates mutational status of 69 potentially oncogenic genes present in the Oncomine Myeloid Research Assay diagnostic panel, specifically, hotspot genes (ABL1, BRAF, CBL, CSFR3, DNMT3A, FLT3, GATA2, HRAS, IDH1, IDH2, JAK2, KIT, KRAS, MOL, MYD88, NPM1, NRAS, PTPN11, SETPB1, SF3B1, SRSF2, U2AF1, WT1), full genes (ASXL1, BCOR, CALR, CEBPA, ETV6, EZH2, IKZF1, NF1, PHF6, PRPF8, RB1, RUNX1, SH2B3, STAG2, TET2, TP53, ZRSR2), and fusion transcripts (ABL1, ALK, BCL2, BRAF, CCDN1, CREBBP, EGFR, ETV6, FGFR1, FGFR2, FUS, HMGA2, JAK2, KMT2A, MECOM, MET, MLLT10, MLLT3, MYBL1, MYH11, NTRK3, NUP214, PDGFRA, PDGFRB, RARA, RBM15, RUNX1, TCF3, TFE3). Only variants with an allelic frequency (VAF) > 5% were reported.

A review of literature on TPO-RA use in the setting of aplastic anemia post-HSCT or thrombocytopenia following chemotherapy for hematologic conditions was performed by searching for indexed articles and published abstracts up to January 2022 in MEDLINE via PubMed and the National Library of Medicine.

## 3. Results

### 3.1. Case Description

A 60-year-old woman was referred to our center to receive autologous stem cell transplant (ASCT) due to amyloidosis AL with renal and cardiac disorder. The latter had been diagnosed one year before, and treated with 4 cycles of bortezomib, cyclophosphamide and dexamethasone without significant improvement. Following stem cell mobilization with granulocyte colony-stimulating factor (G-CSF), 6 × 10^6^ CD34+ cells/kg were collected. After admission, she was conditioned with Melphalan 200 mg/square meter and two days later, she was reinfused with 4 × 10^6^ CD34+ cells/kg. The post-transplant period was complicated by extremely severe acute kidney injury, hemorrhagic shock secondary to hematemesis due to severe mucositis, Pneumococcus pneumonia, sepsis by *Bacillus cereus* and graft failure requiring 3 further peripheral blood stem cell infusions and prolonged stimulation with G-CSF. After 35 days from ASCT, the patient was transfusion-dependent on red blood cells (RBC) and platelets (PLT), and BM evaluation showed severe aplasia. Eltrombopag was started at 50 mg per day (Figure 1), and subsequently increased to 100 mg per day, but then ceased on day 52 due to apparently scarce efficacy on platelet count. Neutrophils’ recovery started from day 45 after ASCT and she was discharged on day 63 with persistent need of RBC and PLT transfusions. On day 91, BM revaluation showed a cellularity of 10% without pathological infiltrate or amyloid deposition. On day 129, due to the persistence of severe thrombocytopenia, a new attempt with eltrombopag at 150 mg per day was performed. The patient progressively became transfusion independent from day 200 for RBC and 249 for PLT and these latter increased above 30 × 10^9^/L from day 262. By day 390, given stable PLT counts > 50 × 10^9^/L, eltrombopag was de-escalated to 100 mg per day and subsequently to 50 mg per day, until discontinuation on day 536. At the last follow up, 30 months after ASCT, amyloidosis is still in remission and the patient displays normal blood counts (Hb 12 g/dL, neutrophils 2070 × 10^6^/L, PLT 151 × 10^9^/L) off therapy. Regarding safety, the patient did not show treatment-emergent adverse events during eltrombopag therapy, including thrombosis and increase in bone marrow reticulin fibrosis (WHO grade MF-0 at all evaluations).

### 3.2. Monitoring of Clonal Evolution during Eltrombopag Therapy

To investigate efficacy and clonal evolution under eltrombopag therapy, BM was performed at 3, 6, and 12 months of eltrombopag treatment. It showed a progressive improvement of cellularity along with an increasing polyclonal lymphoid infiltrate of double-negative T-CD3+ cells (Figure 2). No signs of dysplasia or blasts increase were reported. Cytogenetics was persistently normal. Flow-citometry for paroxysmal nocturnal hemoglobinuria clones was performed at 3 months from eltrombopag start and was negative. By NGS analysis on peripheral blood after 6 months of eltrombopag, a mutation of the RB1 gene, with a variant allele frequency of 52.8%, was detected. A re-evaluation at 12 months did not confirm this finding and showed an absence of mutations.

### 3.3. Cytokine Analysis

Figure 3 shows the cytokine levels during eltrombopag treatment. The T helper1 (Th1) cytokines IFN-γ and TNF-α were at lower limit of normality of controls and remained stable along eltrombopag treatment. The T helper 2 (Th2) cytokines IL-6 and IL-10 levels and the immunoregulatory cytokine IL-17 were higher than in controls, and persisted elevated during therapy, with a peak of IL-6 levels at 12 months. Finally, the inhibitory cytokine TGF-β was progressively increased along eltrombopag treatment, although within the normal range.

### 3.4. Literature Review

Several studies have been published about the use of TPO-RA for aplasia secondary to HSCT and chemotherapy both in adult and pediatric settings (Table 1 and Table 2). A total of 1218 patients received TPO-RA post-HSCT, of whom, 581 (48%) eltrombopag, 122 (10%) romiplostim, and 515 (42%) both agents. The great majority were retrospective studies, including small series of around 30 patients, a larger study of 86 cases, a meta-analysis of 378 patients, and 6 prospective trials (Table 1). 

Overall, they highlight a highly variable efficacy in post HSCT aplasia, and a recent meta-analysis on 378 patients documented a pooled response rate of 73% (95% CI: 68–78%) for both eltrombopag and romiplostim [34]. However, the majority of the studies analyzed the efficacy of these drugs in an allogenic HSCT setting and only few data are available for post ASCT aplasia. To the best of our knowledge, only 8 articles, for a total of 28 cases, reported the use of TPO-RA after ASCT, and pooled efficacy may be calculated from 3 retrospective studies only, including 8 patients with 100% response rate [8,9,10,13,19,27,32,39]. Notably, responding patients within the various reports could discontinue TPO-RA after persistent recovery of cytopenia and bone marrow function. However, such small series carry the bias of reporting only successful cases and a prospective evaluation would be necessary.

Far fewer studies are available about the use of TPO-RA after chemotherapy in hematologic malignancies (2 retrospective, 2 prospective trials, and 1 case report), for a total of 233 patients, the great majority receiving eltrombopag. Different efficacy endpoints were considered, often expressed as median time to PLT count recovery [43,44,45,46,47]. The latter was reported to be about 22.5 (16–43) days in acute myeloid leukemia patients and 7.4 (4.9–9.9) days in lymphoma patients [45,46]. Interestingly, even in these cases, TPO-RA discontinuation was possible after cytopenia recovery. Finally, novel anti-cancer drugs such as immune checkpoint inhibitors (nivolumab, ipilimumab, pembrolizumab, etc.) re-activate the immune system to recognize tumor specific antigens but may in turn cause autoimmune hyperinflammatory reactions, including autoimmune cytopenias. In this context, TPO-RA have been used in 6 cases with a good efficacy and safety profile [48,49,50,51,52,53]. Similarly, CAR-T cell therapy has been related to late-onset cytopenias, resembling primary autoimmune ones, often responding to immunosuppressive drugs [54]. A case report about the use of romiplostim for thrombocytopenia following CAR-T cells in a patient affected by relapsed/refractory diffuse large B-cell lymphoma was recently published [55] (Table 3).

Some ITP patients have been reported to benefit from switching from one TPO-RA to another. However, no data are available in the setting of aplastic anemia post-chemotherapy or after HSCT, except for a case switched from eltrombopag to romiplostim after umbilical stem cell transplant [56].

Regarding safety data, TPO-RA appear to have a good profile also in these off-label settings. From the meta-analysis of Yao et al., the pooled rate of adverse events (AE) was 3% (95%CI: 1–7%), with no severe AEs reported [34]. Similarly, the phase II study conducted by Ahmed S et al. highlighted no differences in grade 2 or higher AE rates between the eltrombopag and placebo arm, particularly regarding bone marrow fibrosis, cataract, vision changes, or thrombosis [39]. Finally, we identified only one case of pulmonary embolism secondary to eltrombopag therapy, while the most frequent reported AE was a transient elevation of liver function tests. 

Research on ClinicalTrials.gov (accessed on 22 January 2022) returned 29 registered trials regarding the use of TPO-RA in aplasia post-chemotherapy for hematologic disease and post-HSCT (Table 4). The results will be important to better define efficacy, dosage, timing, and side effects of these drugs in these off-label settings.

### 3.5. Immunomodulatory Effects of Eltrombopag

Several mechanisms of action have been proposed to explain the immunomodulating properties of eltrombopag. The most agreed hypothesis is the restoration of an antigen-specific tolerance through induction of T cell anergy by increased exposure to PLT antigens [57]. More interesting, eltrombopag appeared to be able to recover immune tolerance by improving T-regulatory cells (T-regs) function and TGF-α levels [58]. The latter are key immunomodulatory factors inducing the suppression of self-reactive clones, but also contributing to bone marrow microenvironment homeostasis. In fact, the TGF-beta superfamily is implied in the regulation of late-stage erythropoiesis that is affected in patients with aplastic anemia and bone marrow failure. These observations are also supported by a recent study by Lozano et al., suggesting a possible role of platelet-derived ectosomes (a type of microparticles) in inducing T-regs differentiation in TPO-RA treated patients, and the involvement of additional cellular pathways (i.e., FOXP3, PTPN1, and PPARγ) that may modulate immune response and induce immune tolerance [57]. Moreover, eltrombopag was shown to restore Fcγ receptor (FcγR) balance through the inhibition of FcγRIIb on monocytes, mitigating phagocytic capacity, often hyperactivated in ITP patients [59]. In this context, the drug may also promote the switch of macrophage phenotype from the M1 proinflammatory type to the M2 anti-inflammatory type, dampening autoinflammation and disease chronicization [60]. Finally, eltrombopag may exert immunomodulating properties by promoting B regulatory cells’ function [61]. An additional mechanism related to autoinflammation and bone marrow failure is the interference of eltrombopag with iron metabolism [62,63,64,65,66]. In fact, the drug exhibits iron-chelating properties that may alleviate bone marrow stress and improve hematopoiesis by avoiding iron-induced damage mechanisms, i.e., reactive oxygen species (ROS) generation. Concurrently, iron chelation could contribute to restoring immune functions, often impaired by disturbance of iron homeostasis [5,67].

The new insights on immunomodulatory effects of eltrombopag are extremely interesting even though most of the current evidence is limited to ITP. Thus, further studies would be important to evaluate these mechanisms in other settings such as primary bone marrow failures (AA and MDS) and aplasia secondary to chemotherapy and HSCT, where the degree of immune activation and the composition of bone marrow microenvironment are different.

## 4. Discussion and Conclusions

We presented a case of prolonged bone marrow aplasia secondary to ASCT treated with eltrombopag at the same dosage used for AA. During treatment, our patient did not show neither hepatologic toxicity nor thrombotic events and bone marrow evaluations did not display an increase in fibrosis. Our observations are consistent with the several evidence about the efficacy and safety of TPO-RA in post-HSCT and aplasia post chemotherapy, where the reported pooled response is about 70%, similarly to primary ITP.

One of the major concerns about the use of TPO-RA, particularly in the oncologic setting, is clonal evolution. In our case, sequential BM evaluation did not show an increase in blasts percentage, nor the emergence of cytogenetic aberrations; furthermore, the detection of RB1 gene mutation by NGS was not confirmed in the subsequent analysis and PNH clone testing was negative. In the literature, the risk of clonal escape is controversially reported for AA and MDS, which, however, carry an intrinsic risk of leukemic evolution [68,69,70]. In this context, a recent trial analyzing the efficacy and safety of eltrombopag added to immunosuppressive therapy in severe AA, also collected sequential mutational data by NGS, and did not detect clonal evolution after a median follow-up of 24 months [71].

Another concern with eltrombopag is the increase in reticulin bone marrow fibrosis. Although specific data in the setting of aplastic anemia following chemotherapy, HSCT, or novel anti-cancer drugs are not available, several studies in ITP and primary aplastic anemia demonstrated that it is mainly low-grade fibrosis, remitting after TPO-RA discontinuation [3,72,73]. Notably, our patient and many of those described in the literature were able to discontinue TPO-RA after robust recovery of cytopenias and bone marrow cellularity. This is also reported for ITP patients in about 1/3 of cases and is likely linked to an immunomodulatory effect of the drug. 

Accordingly, in our study, we demonstrated an immunomodulatory effect of eltrombopag on different cytokines. Particularly, we observed that the patient’s IL-6 and IL-10 levels were increased compared to controls, in agreement with the several reports of increased levels in plasma cell dyscrasias [74,75]; additionally, IL-10 is a negative regulator of Th1 cytokines and might have accounted for the reduced levels of TNF-α and IFN-γ observed during eltrombopag therapy. Consistently, the reduction of the latter two cytokines may have resulted in decreased inhibition of Th 2 response with an amplification of the autoimmune process. Regarding immunoregulatory cytokines, IL-17 was higher than controls, further amplifying the proinflammatory and autoimmune response. Interestingly, the inhibitory cytokine TGF-β, which was initially at the lower limit of normal, increased over time, in an attempt to downregulate the pro-inflammatory/autoimmune response. This might be related with a restoration of T-regs/T-helper 17 cells ratio already described in patients treated with eltrombopag [58]. 

Another interesting finding is the histologic demonstration of a polyclonal double negative T-cells infiltrate in patient bone marrow, increasing over time in parallel with bone marrow cellularity. These cells belong to a neglected class of T cells called CD3+CD4-CD8- and have been described to play a homeostatic role in suppressing exuberant immune responses [76]. Along with the immune reconstitution after ASCT, this observation may further suggest an immunomodulatory effect of eltrombopag in this setting.

Finally, the new insights on the immunomodulatory effects of TPO-RA are intriguingly expanding the spectrum of use of these drugs. In the near future, treatment of “peripheral” (ITP) and “central” (AA, MDS) autoimmune diseases will take advantage of TPO-RA to either stimulate the stem cells compartment and exert a slower, but likely less toxic, immunomodulatory effect.

In conclusion, the case presented and the literature review highlight the efficacy and safety of eltrombopag in aplastic anemia following ASCT, HSCT, and chemotherapy for hematologic malignancies. Furthermore, cytokine studies and bone marrow findings support the emerging immunomodulatory effect of the drug. Altogether, growing evidence suggests eltrombopag use in these off-label settings, particularly in the case of prolonged cytopenias following poor graft function or chemotherapy.

## Figures and Tables

**Figure 1 pharmaceuticals-15-00419-f001:**
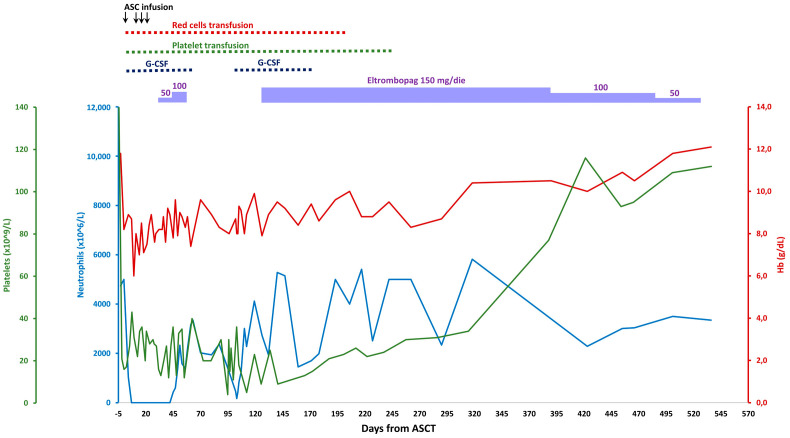
Hemoglobin (red line), neutrophils (blue line) and platelets (green line) trends from ASCT to eltrombopag discontinuation. ASCT, autologous stem cells transplant; G-CSF, granulocyte colony-stimulating factor.

**Figure 2 pharmaceuticals-15-00419-f002:**
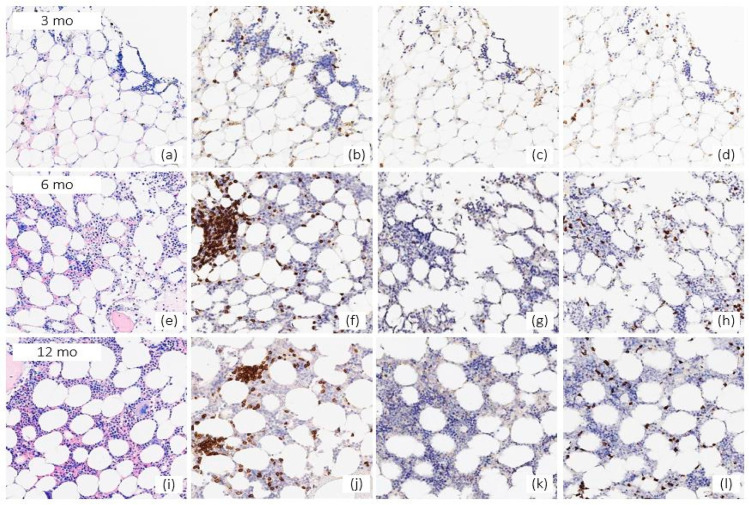
Bone marrow histology at 3, 6 and 12 months (mo) of eltrombopag treatment. The 3-months-biopsy (**a**) Giemsa, 200×, depicts a hypocellular bone marrow, with hematopoiesis mostly represented by erythroid islands, in the absence of remarkable features of morphologic dysplasia, nor increased blast count. An adjoining lymphoid population is very scant and mostly represented by small CD3+ T-cells; (**b**) CD3, 200×, interstitial in their distribution, with very rare CD4+ cells; (**c**) 200× and a minor component of CD8+ cells; (**d**) 200×, thus, consistent with the presence of a CD4-/CD8- subpopulation. The 6-months biopsy documents an increasing cellularity, with relative over-representation of erythroid series; (**e**) Giemsa, 200×, coupled with a heavier T-cell infiltrate 10–15% of overall cellularity; (**f**) CD3, 200×, patterned in a micronodular and interstitial configuration, with no intrasinusoidal features, still partially lacking CD4; (**g**) 200× and CD8; (**h**) 200× expression. The 12-months biopsy displays a hematopoiesis within ranges; (**i**) Giemsa, 200×, notably featuring scattered giant, hyperlobated megakaryocytes (morphologic appearance consistent with therapy related modifications) and a T-cell infiltrate; (**j**) CD3, 200×; (**k**) CD4, 200×; (**l**) CD8, 200× overlapping the previous picture. Grade of marrow fibrosis at 3, 6, and 12 months: MF-0. Not shown in the present panel, no significant amount of lymphoid cells are seen expressing CD56, CD57, TCR-delta, FoxP3, CD25, and TdT. Furthermore, focal, perivascular amyloid deposits are observed, without evidence of residual plasma cell neoplasm component.

**Figure 3 pharmaceuticals-15-00419-f003:**
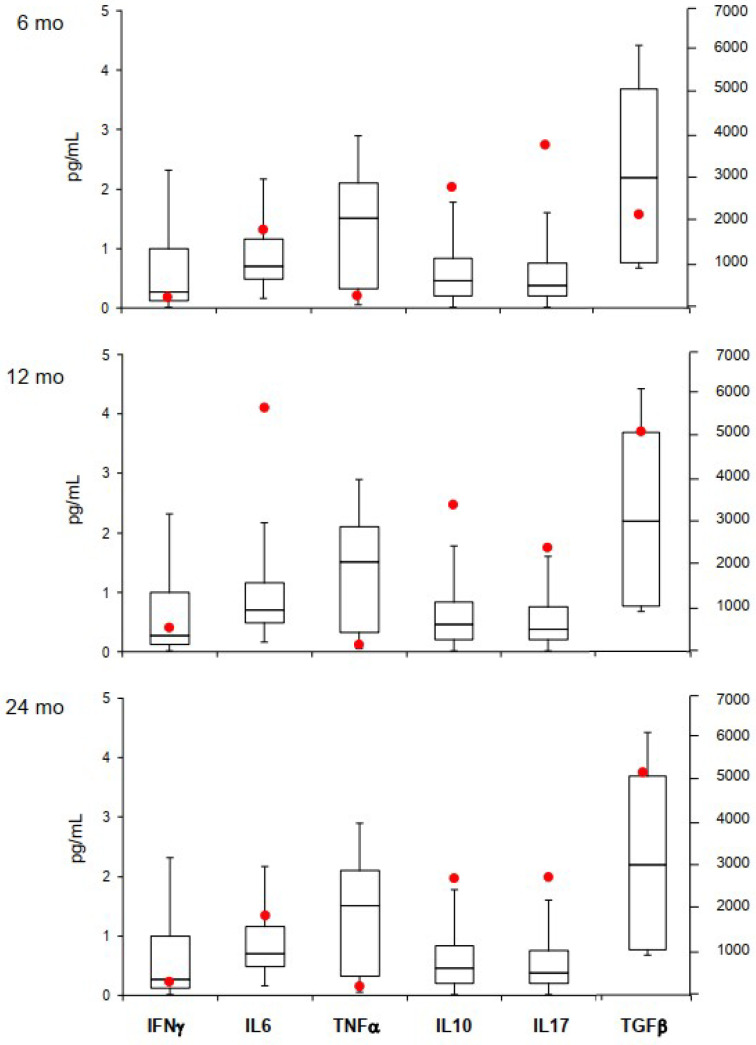
Cytokine levels during eltrombopag treatment. Red dots represent patient’s values; boxplots represent the median, quartiles, and ranges for cytokine levels in 40 age- and sex-matched healthy controls. IFN-γ, interferon gamma; IL, interleukin 6, 10, 17; TGF-β, transforming growth factor beta; TNFα, tumor necrosis factor alpha; 6, 12 and 24 months (mo) of eltrombopag treatment.

**Table 1 pharmaceuticals-15-00419-t001:** Studies on TPO-RA post-HSCT.

Type	No. pts	E/R	Results	References
retrospective	7	R	ORR 100%	Calmettes C et al., Bone Marrow Transpl, 2011 [7]
case report	1	R	ORR 100%	**Gangatharan SA et al., Bone Marrow Transplant, 2011** [8]
case report	2	E	ORR 100%	**Reid R et al., Am J Hematol, 2012** [9]
phase 1 study	19	E	ORR na	**Liesveld JL et al., Biol Blood Marrow Transplant, 2013** [10]
case report	1	E	ORR 100%	Fujimi A et al., Int J Hematol, 2015 [11]
retrospective	8	R	ORR 62.5%	Battipaglia G et al., Bone Marrow Transplant, 2015 [12]
retrospective	12	E	ORR na	**Raut S et al., Indian J Hematol Blood Transf, 2015** [13]
retrospective	12	E	ORR 60%	Tanaka T et al., Biol Blood Marrow Transpl, 2016 [14]
case report	1	E	ORR 100%	Dyba J et al., Transfus Med, 2016 [15]
retrospective	13	R	ORR 53%	Hartranft ME et al., J Oncol Pharm Pract, 2017 [16]
retrospective	12	E	ORR 83%	Tang C et al., J Hematol Oncol, 2018 [17]
retrospective	20	E/R	ORR 60%	Bosch-Vilaseca A et al., Eur J Hematol, 2018 [18]
retrospective	19	E	ORR 74%	**Mori S et al., Biol Blood Marrow Transplant, 2018** [19]
case report	3 (ped)	E	ORR 66.6%	Li S et al., J Pediatr Hematol Oncol, 2019 [20]
retrospective	14	E	ORR 57%	Rivera D et al., Bone Marrow Transpl, 2019 [21]
retrospective	31	E/R	ORR 52%	Rudakova TA et al., Cellular Ther Transplant, 2019 [22]
retrospective	38	E	ORR 63%	Fu H et al., Bone Marrow Transpl, 2019 [23]
retrospective	13	E	ORR 53%	Marotta S et al., Bone Marrow Transpl, 2019 [24]
phase 2 study	13	E	ORR 62%	Yuan C et al., Biol Blood Marrow Transpl, 2019 [25]
retrospective	86	E/R	ORR 72%	Bento L et al., Biol Blood Marrow Transpl, 2019 [26]
retrospective	21	E	ORR 75%	**Samarkandi H et al., Hematol Oncol Stem Cell Ther, 2020** [27]
phase 1/2 study	24	R	ORR 66%	Peffault De Latour R et al., Blood, 2020 [28]
retrospective	9 (ped)	E	ORR 88%	Masetti R et al., Pediatr Blood Cancer, 2020 [29]
retrospective	12	E	ORR 83%	Aydin S et al., Ther Adv Hematol, 2020 [30]
retrospective	32	E	ORR 65.6%	Gao F et al., Ann Hematol, 2020 [31]
review	121/49 E/R	E/R	ORR 67/82% E/R	Mahat U et al., Biol Blood Marrow Transpl, 2020 [32]
retrospective	17	E	ORR 76.5%	Nampoothiri RV et al., Bone Marrow Transplant, 2021 [33]
meta-analysis	378	E/R	ORR 73%	Yao Y et al., Expert Rev Hematol, 2021 [34]
retrospective	18 (ped)	E	ORR 77.7%	Yaman Y et al., Pediatr Transplant, 2021 [35]
retrospective study	5 (ped)	E	ORR 80%	Uria-Oficialdegui ML et al., Pediatr Transplant, 2021 [36]
phase 1 study	20	R	ORR 100%	Christakopoulos GE et al., Transplant Cell Ther, 2021 [37]
phase 2a study	12 (2 ped)	E	ORR na	Pasvolsky O et al., Leuk Lymphoma, 2021 [38]
phase 2 (E vs. PbO)	60 (2/1)	E	ORR 36 vs. 28%	**Ahmed S et al., Transplant Cell Ther, 2021** [39]
retrospective	48	E	ORR 75%	Giammarco S et al., Int J Hematol, 2021 [40]
retrospective	43 (ped)	E	ORR 81%	Qiu KY et al., Br J Clin Pharmacol. 2021 [41]
retrospective	24 (ped)	E	ORR 92%	Liu S et al., Zhonghua Er Ke Za Zhi, 2021 [42]

E, Eltrombopag; HSCT, hemopoietic stem cell transplant; ORR, overall response rate; PbO, placebo; ped, pediatric patients; pts, patients; R, Romiplostim. Bolded studies report the use of TPO-RA for aplasia post-autologous stem cell transplant.

**Table 2 pharmaceuticals-15-00419-t002:** Studies on TPO-RA after chemotherapy.

Type	No. pts	Hematologic Disease	E/R	Results	References
case report	1	MCL	R	ORR 100%	Demeter J et al., Pathol Oncol Res, 2011 [43]
phase 2 study (E vs. PbO)	148 (1/1)	AML	E	na	Frey N et al., Lancet Haematol, 2019 [44]
phase 1 study	14	AML	E	TPR 22.5 (16–43) * d	Strickland SA et al., Leuk Lymphoma, 2020 [45]
retrospective	50	Lymphoma	E	TPR 7.43 + 2.54 ° d	Zhu Q et al., Front Oncol, 2021 [46]
retrospective	20	Lymphoma/MM	R	ORR 10%	Al-Samkari H et al., Haematologica, 2021 [47]

AML, acute myeloid leukemia; d, days; E, Eltrombopag; MCL, mantle cell lymphoma; MM, multiple myeloma; na, not available; ORR, overall response rate; PbO, placebo; pts, patients; R, Romiplostim; TPR, time for platelets count recovery. * median days for platelet recovery (minimum-maximum). ° mean days for platelet recovery (mean + SD).

**Table 3 pharmaceuticals-15-00419-t003:** Studies on TPO-RA after immune checkpoint inhibitors therapy and CAR-T therapy.

Type	Setting	No. pts	Disease	E/R	Results	References
retrospective	CPI (N)	1	melanoma	R	ORR 100%	Kanameishi S et al., Ann Oncol, 2016 [48]
retrospective	CPI (P)	1	NSCLC	E	na	Song P et al., Eur J Cancer, 2019 [49]
retrospective	CPI (P)	1	NSCLC	E	ORR 100%	Ito M et al., Lung Cancer, 2020 [50]
retrospective	CPI (D)	1	SCLC	E	ORR 100%	Suyama T et al., J Clin Exp Hematop, 2021 [51]
retrospective	CPI (N)	1	NSCLC	E	ORR 100%	Fu S et al. J Oncol Pharm Pract, 2021 [52]
retrospective	CPI (N)	1	RCC	E	ORR 0%	Younan RG et al., Allergy Asthma Clin Immunol, 2021 [53]
retrospective	anti-CD19 CAR-T (Axi-cel)	1	NHL	R	ORR 100%	Baur R et al., J Immunother Cancer, 2021 [55]

CPI, checkpoint inhibitors; D, durvalumab; E, Eltrombopag; N, Nivolumab; NHL, Non-Hodgkin Lymphoma; NSCLC, non–small cell lung cancer; ORR, overall response rate; P, Pembrolizumab; pts, patients; R, Romiplostim; RCC, renal cell carcinoma; SCLC, small cell lung cancer.

**Table 4 pharmaceuticals-15-00419-t004:** Registered trials on ClinicalTrials.gov (accessed on 22 January 2022).

Trial No.	Type	Drug	Setting
NCT04673266	Phase 2	R	Post-CT in Lymphoma
NCT04600960	Phase 2	E	Post-CT
NCT04478123	Phase 2	R	Post-CT and post-ASCT in MM, NHL, HL
NCT03948529	Phase 2	E	Post-allogenic HSCT
NCT03902041	Observational	E	Post-allogenic HSCT
NCT03718533	Phase 2	E	Poor graft function
NCT03701217	Phase 2/3	E	Post-CT in AML
NCT03603795	Phase 2	E	Post-CT in AML of elderly pts
NCT03515096	Phase 3	E vs. rhTPO	Post-HSCT
NCT03437603	Phase 2	E	Post-allogenic HSCT
NCT03343847	Phase 3	R	Post-CT in Lymphoma
NCT02071901	Phase 2	E	Post-CT in AML of elderly pts
NCT02052882	Phase 2	R	Post-CT
NCT02046291	Phase 1	R	Post-UCBT
NCT01980030	Phase 1/2	R	Post-allogenic HSCT
NCT01940562	Phase 2	E	Post-UCBT in pediatric pts
NCT01927731	Phase 2	E	Post-HSCT
NCT01791101	Phase 2	E	Post-HSCT
NCT01757145	Phase 2	E	Post-UCBT
NCT01676961	Phase 2	R	Post-CT in MM
NCT01656252	Phase 1/2	E	During CT in AML
NCT01516619	Phase 2	R	Post-CT in NHL
NCT01428635	Phase 2/3	E	During TKI in CML and PMF
NCT01286675	Early phase 1	E & G-CSF	Mobilization CD34+ cell in MM
NCT01000051	Phase 2	E	Post-HSCT
NCT00903929	Phase 1	E	Post-TBI
NCT00299182	Phase 1/2	R	Post-CT in Lymphoma
NCT00283439	Phase 1/2	R	Post-CT in Lymphoma
NCT00102726	Phase 2	E	Post-CT

AML, acute myeloid leukemia; CML, chronic myeloid leukemia; CT, chemotherapy; E, Eltrombopag; HL, Hodgkin lymphoma; HSCT, hematopoietic stem cell transplant; MM, multiple myeloma; NHL, non-Hodgkin lymphoma; PMF, primary myelofibrosis; R, Romiplostim; TBI, total body irradiation; TKI, tyrosine kinase inhibitor; UCBT, umbilical cord blood transplant.

## Data Availability

Data is contained within the article.

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
