# Peer review of "Efficacy and Immunomodulating Properties of Eltrombopag in Aplastic Anemia following Autologous Stem Cell Transplant: Case Report and Review of the Literature"

_pharmaceuticals, 2022, doi:10.3390/ph15040419_

Round 1

Reviewer 1 Report

Authors reported a case with aplastic anemia post-HSCT treated with TPR-RA and reviewed the literature on TPO-RA use in the setting of aplastic anemia post-HSCT or 67 thrombocytopenia following chemotherapy for hematologic conditions.

Although this manuscript is potentially interesting, several issues arise.

1. Authors should show the attractive points in this case.

2. Authors should show “take home message”.

3.Serum TPO levels may be useful.

4.Figure 3 should be explained in legends. Is Figure 3 important?

5.Authors should analyze the results of review.

6.If possible, authors should statistically analyze the results of review.

7.Some schema may be attractive.

Author Response

Reviewer 1

Authors reported a case with aplastic anemia post-HSCT treated with TPR-RA and reviewed the literature on TPO-RA use in the setting of aplastic anemia post-HSCT or 67 thrombocytopenia following chemotherapy for hematologic conditions.

We thank the Referee for the careful revision.

Although this manuscript is potentially interesting, several issues arise.

1. Authors should show the attractive points in this case.

2. Authors should show “take home message”.

Our manuscript describes the efficacy of eltrombopag in the setting of aplastic anemia following autologous stem cell transplant. This is a poorly explored use of eltrombopag, in a condition characterized by a strongly altered immunologic milieu, and offered the unique occasion to further assess the immunomodulatory effect of the drug. A further attractive point is the review of the literature that improves our knowledge about eltrombopag use in off label settings. These points and the main “take home messages” have now been added to the conclusions:

“In conclusion, the case presented and the literature review highlight the efficacy and safety of eltrombopag in aplastic anemia following ASCT, HSCT, and chemotherapy for hematologic malignancies. Furthermore, cytokine studies and bone marrow findings support the emerging immunomodulatory effect of the drug. Altogether, growing evidence suggest eltrombopag use in these off-label settings.”

3.Serum TPO levels may be useful.

As a matter of fact, we did not test serum TPO levels in our patients. Although we agree with the Referee that TPO levels may be speculatively useful in this case, they are not routinely measured before TPO-RA treatment. Moreover, data about the predictive role of TPO levels in patients receiving TPO-RA from the literature are controversial.

4.Figure 3 should be explained in legends. Is Figure 3 important?

We added the title and legend of Figure 3. We believe that the figure adds some value to the manuscript and completes the description of the text. If possible, we would like to keep it.

“Figure 3. Cytokine levels during eltrombopag treatment. IFN-γ interferon gamma; IL interleukin 6, 10, 17; TGF-β, transforming growth factor beta; TNFα, tumor necrosis factor alpha; 6, 12 and 24 months (mo) of eltrombopag treatment.”

5.Authors should analyze the results of review. 6.If possible, authors should statistically analyze the results of review. 7.Some schema may be attractive.

We agree that the Results of the literature review are interesting and important. As a matter of fact, we produced 4 detailed tables depicting TPO-RA use post HSCT, chemotherapy, novel drugs, and ongoing clinical trials. In the text we chose to summarize the data obtained to draw a message for the Reader. Since this is not a meta-analysis nor a systematic review, a definite statistical analysis was not feasible. However, we tried to pool similar studies to calculate the frequency of response to TPO-RA in the various settings. Finally, the article is already crowded with 4 tables and 3 figures so that we did not consider to add additional schemas. However, if the Referee would suggest a specific one, we will happy to consider it.

Reviewer 2 Report

The authors in this manuscript presented an interesting and important case report regarding a 60-year-old woman having amyloidosis AL with the renal and cardiac disorder, who was receiving autologous stem cell transplant (ASCT) and prolonged treatments with Eltrombopag, a thrombopoietin receptor agonist, for the severe immune thrombocytopenia and aplastic anemia after ASCT. Upon prolonged treatment of Eltrombopag, the patient progressively became transfusion-independent for the production of RBCs and platelets. After 536 days, the treatment of Eltrombopag was discontinued and 30 months after ASCT when the last follow-up was prescribed, amyloidosis did not come back again. The authors found that treating the patient with Eltrombopag for the aplastic anemia and thrombocytopenia and the effects went beyond the TPO-receptor stimulation and pointed at immunomodulating properties of these drugs.

Overall, the manuscript was well written with the aids of schematic illustration, figure presentation, and all the tables for the literature review listing how patients effectively and safely responded to TPO-RA. This case report should be able to provide scientists and oncologists an excellent example of how amyloidosis patients can be treated with Eltrombopag to prevent anaplastic anemia and thrombocytopenia.

Author Response

Reviewer 2

The authors in this manuscript presented an interesting and important case report regarding a 60-year-old woman having amyloidosis AL with the renal and cardiac disorder, who was receiving autologous stem cell transplant (ASCT) and prolonged treatments with Eltrombopag, a thrombopoietin receptor agonist, for the severe immune thrombocytopenia and aplastic anemia after ASCT. Upon prolonged treatment of Eltrombopag, the patient progressively became transfusion-independent for the production of RBCs and platelets. After 536 days, the treatment of Eltrombopag was discontinued and 30 months after ASCT when the last follow-up was prescribed, amyloidosis did not come back again. The authors found that treating the patient with Eltrombopag for the aplastic anemia and thrombocytopenia and the effects went beyond the TPO-receptor stimulation and pointed at immunomodulating properties of these drugs.

Overall, the manuscript was well written with the aids of schematic illustration, figure presentation, and all the tables for the literature review listing how patients effectively and safely responded to TPO-RA. This case report should be able to provide scientists and oncologists an excellent example of how amyloidosis patients can be treated with Eltrombopag to prevent anaplastic anemia and thrombocytopenia.

We would like to sincerely thank the Referee for the revision and for the positive comments and feedback. We are happy that you enjoyed our manuscript.

Reviewer 3 Report

The aim of the present study was to assess the efficacy and the immunomodulating effect of eltrombopag, a TPO-mimetic agent, in the setting of bone marrow aplasia following chemotherapy and autologous stem cells transplantation. Furthermore, a review of the current literature on the topic is provided. Due to the lack of therapeutic options in this setting, the topic is of clinical interest; the report is very well written, and the discussion is well developed.

minor points

  1. Bone marrow fibrosis is one of the major concerns about using eltrombopag for long periods of time. The authors mention in the discussion that the patient did not display an increase in fibrosis. I would mention this relevant aspect also in the clinical case description, grading BM fibrosis at diagnosis and follow-up. Possibly, data on fibrosis with TPO mimetics could be further discussed.

  2. The description of the adverse events experienced during treatment should be mentioned in the clinical case part and not mentioned for the first time in the discussion (line 242).

  3. Is there any data describing a gain of responses in switching from a TPO mimetic to another (for example from eltrombopag to romiplostim), as it is described in ITP? This aspect could also be mentioned in the discussion

Author Response

Reviewer 3

The aim of the present study was to assess the efficacy and the immunomodulating effect of eltrombopag, a TPO-mimetic agent, in the setting of bone marrow aplasia following chemotherapy and autologous stem cells transplantation. Furthermore, a review of the current literature on the topic is provided. Due to the lack of therapeutic options in this setting, the topic is of clinical interest; the report is very well written, and the discussion is well developed.

We would like to thank the Referee for reviewing our manuscript and for the helpful suggestions.

minor points

  1. Bone marrow fibrosis is one of the major concerns about using eltrombopag for long periods of time. The authors mention in the discussion that the patient did not display an increase in fibrosis. I would mention this relevant aspect also in the clinical case description, grading BM fibrosis at diagnosis and follow-up. Possibly, data on fibrosis with TPO mimetics could be further discussed.

We agree with the Referee:

  • we added a sentence to the results:

“Regarding safety, the patient did not show treatment emergent adverse events during eltrombopag therapy, including thrombosis and increase in bone marrow reticulin fibrosis (WHO grade MF-0 at all evaluations).”

  • we added a sentence in the legend of Figure 2:

“Grade of marrow fibrosis at 3, 6 and 12 months: MF-0.”

  • we added a paragraph to the discussion:

“Another concern with eltrombopag, is the increase in reticulin bone marrow fibrosis. Although specific data in the setting of aplastic anemia following chemotherapy, HSCT or novel anti-cancer drugs are not available, several studies in ITP and primary aplastic anemia demonstrated that it is mainly low grade fibrosis, remitting after TPO-RA discontinuation [71-73].”

Ref:

  1. Wong RSM et al. Safety and efficacy of long-term treatment of chronic/persistent ITP with eltrombopag: final results of the EXTEND study. Blood. 2017 Dec 7;130(23):2527-2536.
  2. Olnes MJ et al. Eltrombopag and improved hematopoiesis in refractory aplastic anemia. N Engl J Med. 2012 Jul 5;367(1):11-9.
  3. Ghanima W et al. Thrombopoietin receptor agonists: ten years later. Haematologica 2018 Volume 104(6):1112-1123.

  1. The description of the adverse events experienced during treatment should be mentioned in the clinical case part and not mentioned for the first time in the discussion (line 242).

We agree with the Referee and added a sentence to the results: Regarding safety, the patient did not show treatment emergent adverse events during eltrombopag therapy, including thrombosis and increase in bone marrow reticulin fibrosis (WHO grade MF-0 at all evaluations).”

  1. Is there any data describing a gain of responses in switching from a TPO mimetic to another (for example from eltrombopag to romiplostim), as it is described in ITP? This aspect could also be mentioned in the discussion

To the best of our knowledge, no data exist about the switch from eltrombopag to romiplostim or vice versa in the setting of aplastic anemia post chemotherapy/CPI, HSCT or ASCT.

From a revision of the literature, excluding the articles about the TPO-RA switch in the ITP setting, we found only a study about the switch from eltrombopag to romiplostim in primary aplastic anemia (Ise M et al. Romiplostim is effective for eltrombopag-refractory aplastic anemia: results of a retrospective study. Int J Hematol. 2020 Dec;112(6):787-794) and a case report about the switch from eltrombopag to romiplostim in a case of ITP developed after umbilical cord blood transplant (Matsumoto R et al. Acute myeloid leukemia developing secondary immune thrombocytopenia after umbilical cord blood transplantation. Rinsho Ketsueki. 2017;58(5):433-437).

We added a sentence and a reference to the “Literature Review” paragraph:

Some ITP patients have been reported to benefit from switching from one TPO-RA to another. However, no data are available in the setting of aplastic anemia post-chemotherapy or after HSCT, except for a case switched from eltrombopag to romiplostim after umbilical stem cell transplant [Matsumoto R et al. Acute myeloid leukemia developing secondary immune thrombocytopenia after umbilical cord blood transplantation. Rinsho Ketsueki. 2017;58(5):433-437)].

Reviewer 4 Report

The authors describe the case of a patient with aplastic anemia following autologous stem cell transplant treated with eltrombopag, reviewing also literature data about this topic. I suggest some additions.

In table 1, please add:

  • type of HSCT (autologous or allogenic);
  • type of hematologic disease, before HSCT;

In table 3, please add:

  • type of hematologic disease;
  • checkpoint inhibitor used;
  • CAR-T cells used.

Author Response

Reviewer 4

The authors describe the case of a patient with aplastic anemia following autologous stem cell transplant treated with eltrombopag, reviewing also literature data about this topic. I suggest some additions.

We would like to thank the Referee for the careful revision our manuscript and for the ameliorating suggestions.

In table 1, please add:

  • type of HSCT (autologous or allogenic);

We added the type of HSCT in the table 1 legend Bolded studies report the use of TPO-RA for aplasia post-autologous stem cell transplant.

  • type of hematologic disease, before HSCT

The types of hematologic diseases are unfortunately several since most studies report case series (i.e., 7 to 121 patients in each study). Therefore, we could not add these many data to the tables (it would be many diseases in a single row).

In table 3, please add:

  • type of hematologic disease;
  • checkpoint inhibitor used;
  • CAR-T cells used.

As suggested, we add the type of hematologic disease, the CPI and the CAR-T types.

Round 2

Reviewer 1 Report

Authors partially improved revised manuscript. However, several issues arrive.

  1. What did boxplots show? Pleases explain in the legend.
  2. Did author recommend all patients with transplantation? Authors should show the criteria for administration of Eltrombopag?
  3. Are there contraindication for Eltrombopag?
  4. Table 1 What is “ped” and “p”?
  5. Table 2 What is “na” and “d”?
  6. What is (16-43) and (4.9-9.9)? range ? 25-75%tile? 2.5-97.5%tile?
  7. Further statistical analysis may be possible for phase 2 trial.

Author Response

Dear Editor,
Please find enclosed the revised version of our article after the second round of revision. Major changes 
have been marked as requested.
Here follow the point-by-point answers to Reviewers’ suggestions.
We hope that you will find our revised manuscript suitable for publication in Pharmaceuticals.
Best regards
Marta Bortolotti
Reviewer 1
Authors partially improved revised manuscript. However, several issues arrive.
We thank the Referee for the careful revision.
1. What did boxplots show? Pleases explain in the legend.
Boxplots represent the median, quartiles and ranges for age and sex matched healthy controls. This has 
been now added to figure legend.
“Cytokine levels during eltrombopag treatment. Red dots represent patient’s values; boxplots represent 
the median, quartiles and ranges for cytokine levels in 40 age and sex matched healthy controls.”
2. Did author recommend all patients with transplantation? Authors should show the criteria for 
administration of Eltrombopag?
We agree with the Referee that it would be nice to give straightforward indications for eltrombopag 
therapy. However, there are no clearcut indications for these off-label settings. Basing on published 
literature, it seems that eltrombopag may be beneficial in patients displaying persistent cytopenia due to 
poor graft function (i.e., delayed engraftment), or prolonged hypoproductive cytopenia following 
chemotherapy. Furthermore, we noticed that in these setting, the recovery of cytopenias and bone 
marrow function may lead to TPO-RA discontinuation (i.e., treatment free remission) in most of cases, 
including ours. We updated the literature review and the conclusions accordingly.
Literature review:
“Notably, responding patients within the various reports could discontinue TPO-RA after persistent 
recovery of cytopenia and bone marrow function”
Discussion and conclusions
“Notably, our patient and many of those described in the literature, were able to dis-continue TPO-RA 
after robust recovery of cytopenias and bone marrow cellularity. This is also reported for ITP patients in 
about 1/3 of cases and is likely linked to an im-munomodulatory effect of the drug.”
“In conclusion, the case presented and the literature review highlight the efficacy and safety of 
eltrombopag in aplastic anemia following ASCT, HSCT, and chemotherapy for hematologic malignancies. 
Furthermore, cytokine studies and bone marrow findings support the emerging immunomodulatory 
effect of the drug. Altogether, growing evidence suggest eltrombopag use in these off-label settings, 
particularly in case of prolonged cytopenias following poor graft function or chemotherapy.”
3. Are there contraindication for Eltrombopag?
At present, there are no clearcut contraindication for eltrombopag treatment. The main warnings reside 
in the BM fibrosis, the thrombotic risk and that of clonal evolution, that have already been discussed in 
the “Discussion and conclusions” section.
4. Table 1 What is “ped” and “p”?
5. Table 2 What is “na” and “d”?
We apologize for the missing explanations of the abbreviations and we added them to the table legends.
6. What is (16-43) and (4.9-9.9)? range? 25-75%tile? 2.5-97.5%tile?
We corrected the table and table legend accordingly. In the first case “22.5(16-43)” represent the median
time to platelet recovery in days (range). In the second case it was reported as mean plus/minus
standard deviation (SD), but we tried to simplify it as median (range) to increase results homogeneity. As 
suggested by the Referee we now put the original data: mean + SD (7,43 + 2,54) days [Zhu Q et al, 46].
7. Further statistical analysis may be possible for phase 2 trial.
Thank you. We believe that a statistical analysis of different studies is more indicated for metanalysis and 
systematic reviews and is beyond the scope of the current article. In any case, we cited an emeritus 
metanalysis of 378 cases of post-HSCT aplasia treated with eltrombopag and romiplostim and the 
relative results. We are however open to eventual specific recommendations by the Reviewer.

Reviewer 2 Report

I have no further suggestion.

Author Response

We thank the Referee.

Reviewer 3 Report

the authors exhaustively adressed the points I raised

Author Response

We thank the Referee.

Reviewer 4 Report

Ok.

Author Response

We thank the Referee.

Round 3

Reviewer 1 Report

Authors sufficiently responded to all comments.